# Position: Uncertainty is a Strategic Signal in Human–AI Decision Making

Achref Doula [1 2]   Otthein Herzog [2 3]   Siegfried Zhiqiang Wu [2]   Max Mühlhäuser [1 2]

## Abstract

AI-assisted decision-making is subject to AI model uncertainty. Prior work has proposed to make this uncertainty explicit to increase trust and transparency, but its behavioral role was rarely studied. This position paper argues, from a game-theoretic perspective, that human–AI decision support should be viewed as a repeated mechanism in which AI uncertainty functions as a strategic signal that shapes how users adopt reliance policies over time. We formalize a framework in which the interface specifies uncertainty signals, user responses such as accepting versus verifying, and the resulting policy-shaping consequences. These repeated steps are used to characterize near-separating reliance regimes. A first pilot study conducted with 180 participants supports our proposition: Our game-theoretic mechanism increased verification and sharply reduced blind acceptance of wrong AI outputs. These initial results support treating human–AI interaction as a game-theoretic mechanism with uncertainty as a strategic signal, rather than a static model property or purely informational label.

## 1. Introduction

Uncertainty has become a central concern as AI systems move from offline prediction to deployed decision aids in domains such as medicine, autonomous driving (Doula et al., 2023), and finance (Amann et al., 2020). In these settings, people repeatedly choose whether to rely on model outputs or verify them, and many modern systems, such as deep learning models and LLM-based assistants, surface an uncertainty or confidence signal, via scores, bands, or traffic-light labels, to foster reliability, trust, and transparency (Kay et al., 2016; Padilla et al., 2022). A common implicit pipeline underlies many designs: if a model produces uncertainty estimates and these are displayed in a clear interface, then users will naturally translate them into selective reliance, treating high-uncertainty outputs with caution and low-uncertainty outputs as safe to follow (Vodrahalli et al., 2022). Much of the surrounding work evaluates uncertainty at this level, asking whether estimates are numerically well behaved and whether visual encodings are interpretable (Guo et al., 2017; Lakshminarayanan et al., 2017; Padilla et al., 2022), and there is evidence that such displays can support more selective reliance in some settings.

However, studies of human–AI decision making repeatedly show that this pipeline is fragile and often only partially realized in practice. Users frequently treat AI recommendations as default answers, exhibit both overreliance and underreliance, and do not consistently adjust their behavior in response to displayed uncertainty or explanations (Buçinca et al., 2021; Vasconcelos et al., 2023). In some domains, AI assistance yields limited or even negative improvements relative to unaided performance, particularly when incorrect recommendations are easy to accept and costly to detect (Buçinca et al., 2021). This body of evidence suggests that simply surfacing uncertainty, while often helpful, is not enough: human responses are shaped by cognitive effort, perceived costs and benefits of checking, evolving beliefs about the system, and the ambient incentives provided by the task and organization (Vasconcelos et al., 2023; Buçinca et al., 2021).

We argue that this is not just an empirical annoyance but a conceptual gap. In deployed human–AI systems, uncertainty is not merely a label attached to a prediction; it is part of a control channel through which the model can influence human actions. The impact of that channel depends on how signals interact with incentives, verification costs, error asymmetries, and limited attention. In particular, if uncertainty indicators are added as neutral annotations that do not affect interaction structure or its consequences, they can behave like cheap talk: they change what is shown on the screen without necessarily changing what people do (Crawford & Sobel, 1982). Our stance is complementary to work on uncertainty estimation and communication but distinct in emphasis (Guo et al., 2017; Lakshminarayanan et al., 2017; Kendall & Gal, 2017; Padilla et al., 2022). We do not question the importance of interpretable uncertainty

[1]Technical University of Darmstadt, Darmstadt, Germany [2]Tongji University, Shanghai, China [3] University of Bremen, Bremen, Germany. Correspondence to: Achref Doula <achref.doula@gmail.com>.

*Proceedings of the 43$^{rd}$ International Conference on Machine Learning*, Seoul, South Korea. PMLR 306, 2026. Copyright 2026 by the author(s).

displays and the benefits they can already bring; instead, we ask how such displays interact with incentives, feedback, and repeated use to shape *policies of reliance*. **Our position is that human–AI decision support should be treated, from a game-theoretic perspective, as a repeated mechanism in which uncertainty functions as a strategic signal and design lever that affects how users adopt reliance policies over time, rather than as a static model property or purely informational label.** By adopting this lens, we shift attention from the static encoding of uncertainty to the *mechanism* in which it is embedded: what actions are supported, what feedback follows, and how consequences accumulate across rounds.

Viewed this way, the central question is no longer whether uncertainty can be displayed, but how an uncertainty signal interacts with verification costs, asymmetric errors, incentives, and feedback to induce stable *policies of reliance* over repeated use. To analyze this interaction, we adopt a game-theoretic perspective: it provides a compact language for strategic signaling, costly action (e.g., verification), and adaptation over repeated rounds. In our framework, an AI assistant emits a recommendation together with an uncertainty signal, and a human chooses whether to accept the recommendation or incur effort to verify it. The interface specifies the interaction structure: what signals can be expressed, what actions users can take, and what feedback they receive after each decision. It also determines how decision consequences accumulate across rounds (e.g., accuracy, and the costs of mistakes), which we summarize abstractly as *payoffs*. Our focus is on repeated interaction: users observe outcomes, update beliefs about the assistant and its signals, and gradually adopt heuristics about when to follow or verify. We do not claim that all human–AI interaction is literally a game in the strict formal sense; rather, this lens makes explicit how uncertainty, incentives, and interface structure jointly shape reliance patterns.

We provide initial empirical support for this perspective with a between-subjects study using a multiple-choice question task. In our pilot study, 180 participants answered 20 questions spanning multiple topics under one of three mechanisms: a human-only baseline with no AI, an uncertainty-only interface where an AI recommendation and uncertainty banner were shown but no per-item feedback or scoring was provided, and a game-theoretic interface that added per-item feedback and a simple scoring rule. The game-theoretic mechanism increased verification rates, reduced blind following and harmful misuse (accepting incorrect recommendations without checking), and yielded statistically significant accuracy gains relative to both the human-only and uncertainty-only conditions, while further sharpening how participants condition their verification behavior on the AI's uncertainty signal.

## 2. Background and Related Work

We briefly review three strands that are most relevant for our position: (i) uncertainty in machine learning and decision support, (ii) empirical studies of human reliance on AI assistance, and (iii) game-theoretic and mechanism-design perspectives on information and delegation (Bhatt et al., 2021; Kleinberg et al., 2018; Green & Chen, 2019). Across these areas, uncertainty is rarely treated as a strategic signal within a repeated mechanism that shapes users' learned reliance policies, which is the gap we aim to address.

### 2.1. Uncertainty in AI and Decision Support

A large body of research develops methods to quantify predictive uncertainty in learned models, including Bayesian approaches, ensemble methods, and post-hoc uncertainty scoring (Gal & Ghahramani, 2016; Kendall & Gal, 2017; Lakshminarayanan et al., 2017; Guo et al., 2017). These techniques are often evaluated in terms of statistical properties or downstream metrics such as risk-sensitive loss and selective prediction, where models may abstain or route difficult cases elsewhere (Geifman & El-Yaniv, 2017; 2019; Xin et al., 2021). In many decision-support applications, such uncertainty estimates are surfaced directly to end users, for instance as confidence scores, probability bars, or discrete labels, with the expectation that people will use them to decide when to trust the model (Bhatt et al., 2021; Kay et al., 2016; Padilla et al., 2022).

This line of work establishes that uncertainty can be computed and that, in principle, decisions that condition on it can improve performance. However, it typically abstracts away from the interaction mechanism: how uncertainty is embedded in the user interface, what actions it supports (e.g., acceptance versus verification), and what feedback users receive about the consequences of their choices over time (Bhatt et al., 2021; Geifman & El-Yaniv, 2017; Xin et al., 2021). In this formulation, the human decision maker is often treated as an idealized agent who correctly interprets and optimally acts on uncertainty information. Our perspective instead focuses on how real users, operating under cognitive and organizational constraints, adapt their reliance behavior within specific mechanisms.

### 2.2. Human Reliance on AI

Empirical studies of human–AI decision making paint a complex picture of assistance. Across domains such as medical triage, assisted driving (Doula et al., 2024), and emergency intervention (Doula et al., 2022), people frequently exhibit both overreliance and underreliance: they may accept incorrect recommendations without checking, or override correct recommendations unnecessarily. Experiments on explanations, confidence displays, and related

decision aids show that adding information does not guarantee better outcomes; users may ignore uncertainty indicators, misinterpret them, or treat model suggestions as defaults regardless of their stated reliability (Bansal et al., 2021; Buçinca et al., 2021; Zhang et al., 2020; Lai & Tan, 2019; Green & Chen, 2019; 2021; Alon-Barkat & Busuioc, 2023; Vasconcelos et al., 2023; Zhang et al., 2024; Buçinca et al., 2026). Behavior is shaped by perceived costs of verification, time pressure, prior beliefs about automation, and institutional incentives, not only by what information is available (Vasconcelos et al., 2023; Greiner et al., 2026).

A parallel line of work studies how to *communicate* uncertainty to end users, comparing numeric probabilities, intervals, verbal qualifiers, and graphical or frequency-based formats, and measuring effects on comprehension, perceived transparency, and decision quality (Kay et al., 2016; Padilla et al., 2022; Van Der Bles et al., 2019; Lammers et al., 2024). There, uncertainty is typically an exogenous quantity to be faithfully conveyed, often evaluated in one-shot or short-horizon tasks. Our stance is complementary but distinct: we focus on how uncertainty interacts with incentives and feedback to shape *policies of reliance*, shifting attention from static displays to *mechanisms*.

### 2.3. Game-Theoretic Design Perspectives

Game theory and mechanism design provide tools for analyzing communication and action under incentives and costs. Work on signaling, cheap talk, and delegation characterizes when informative communication is sustained versus when it collapses into pooling, and how commitment and action costs shape equilibrium behavior (Crawford & Sobel, 1982; Farrell & Rabin, 1996). In information design, these frameworks relate signal and consequence structure to behavioral outcomes (Kamenica & Gentzkow, 2011).

Recent work has imported incentive-aware reasoning into algorithmic decision making (e.g., incentive-aware prediction and strategic classification) (Kleinberg et al., 2018; Green & Chen, 2019; 2021; Noti & Chen, 2023; Alur et al., 2024; Freisinger & Schneider, 2025; Greiner et al., 2026). Most of this literature studies strategic behavior by data providers or decision subjects rather than the end user interacting repeatedly with an AI assistant. Yet, the latter setting is precisely where game-theoretic design tools are natural: an assistant sends an uncertainty signal, the user chooses a costly response (follow vs. verify), and feedback/incentives shape future reliance. We build on these perspectives to model uncertainty displays as a mechanism-level design lever in this repeated user–assistant loop.

## 3. A Game-Theoretic Framework for Repeated Human–AI Interaction

We now make our position more precise by introducing a game-theoretic framework for repeated human–AI interaction. The goal is not to model all psychological or organizational details, but to isolate structural components that determine how uncertainty influences reliance: the signals the assistant may send, the actions available to the human, the feedback revealed after each decision, and the payoffs that accrue over time. In this framework, different interfaces appear as different mechanisms, and uncertainty appears as a signal whose effect is mediated by the mechanism.

Formally, the framework can be viewed as a repeated *signaling game* between an AI assistant (sender) and a human decision maker (receiver). The assistant observes information about an underlying state and emits a signal together with a recommendation; the human observes the signal and chooses an action; and the mechanism determines what feedback and payoffs result. Our position is twofold: (i) in human–AI systems, uncertainty should be engineered and evaluated at the level of the mechanism rather than as a property of a model in isolation; and (ii) standard signaling-game categories, such as pooling and separating regimes, provide a useful vocabulary for describing and targeting patterns of human reliance.

### 3.1. Interaction Protocol and Mechanism

We consider a repeated interaction between a human and an AI assistant over rounds $t = 1, \ldots, T$. In round $t$, a decision case $x_t \in \mathcal{X}$ (the user-visible context, e.g., a question with answer options) is presented, with an associated correct outcome $y_t \in \mathcal{Y}$ that may or may not be known. The assistant observes $x_t$ and its internal state (e.g., a posterior probability over $\mathcal{Y}$) and produces a recommendation together with an *uncertainty signal* (e.g., an uncertainty score or a HIGH/LOW uncertainty banner), acting as the sender in a signaling game. Thus, an interaction in round $t$ proceeds as:

1. The assistant outputs a recommendation $\hat{y}_t \in \mathcal{Y}$ and an uncertainty signal $s_t \in \mathcal{S}$.

2. The human observes $(x_t, \hat{y}_t, s_t)$ and chooses an action $a_t \in \mathcal{A}$ (e.g., accept vs. verify).

3. An outcome is realized: the environment may reveal $y_t$, and the mechanism assigns a payoff $r_t = R(a_t, \hat{y}_t, y_t)$ according to a payoff function $R$.

A *mechanism* is a tuple $M = (\mathcal{S}, \mathcal{A}, F, R)$, where $\mathcal{S}$ is the signal space, $\mathcal{A}$ the action space, $F$ specifies what feedback is revealed after each round (e.g., whether $y_t$ and the assistant's correctness are observed), and $R$ maps actions and outcomes to payoffs. The assistant's mapping $x_t \mapsto (\hat{y}_t, s_t)$

is the sender strategy, determined by the model and a signaling rule (e.g., thresholding an uncertainty score into HIGH/LOW). The mechanism $M$ plays the role of the game form, determining how signals and actions translate into feedback and payoffs across rounds.

The mechanism $M$ is fixed over rounds, but the human can adapt. Let $h_t$ denote the history up to round $t$ (past signals, actions, feedback, and payoffs). A *reliance policy* is a receiver strategy

$$\pi : (\mathcal{X} \times \mathcal{Y} \times \mathcal{S} \times \mathcal{H}) \to \mathcal{A}, \qquad a_t = \pi(x_t, \hat{y}_t, s_t, h_t),$$

where $\mathcal{H}$ is the space of possible histories. In practice, $\pi$ may be implemented by simple heuristics (e.g., "accept when uncertainty is low, otherwise verify") rather than fully optimal planning.

Given a sender strategy and a mechanism, a stationary reliance policy $\pi$ together with beliefs about correctness conditional on signals can be interpreted as the receiver component of a steady-state Bayesian Nash equilibrium of the repeated signaling game. We will not characterize equilibria; instead, we use pooling and (near-)separating regimes to organize the qualitative behaviors that different mechanisms tend to support.

### 3.2. Uncertainty as a Signal in a Signaling Game

Within this framework, what makes a signal "uncertainty" is not its semantics but its statistical relation to error. For a fixed assistant and mechanism, let

$$q(s) \;=\; \Pr\big(y_t = \hat{y}_t \mid s_t = s\big)$$

denote the human's (possibly subjective) belief about the correctness of the recommendation given signal $s$. In many deployments, correctness is not observable at decision time, so $q(s)$ can only be learned from whatever (often sparse) feedback the system surfaces, e.g., audited or probe cases. In our binary uncertainty banner, HIGH denotes higher uncertainty than LOW; when the assistant's internal scores are informative about error and the mapping to $\mathcal{S}$ is monotone, one therefore expects $q(\text{LOW}) > q(\text{HIGH})$, but the definition applies to arbitrary signal spaces.

The signal $s_t$ is *strategic* in the sense that the human's policy is allowed to depend on it. At the signal level, a policy can be summarized by

$$\alpha(s) = \Pr(a_t = \text{ACCEPT} \mid s_t = s),$$

with $1 - \alpha(s)$ the corresponding verification probability. In a signaling-game view, $\alpha$ is part of the receiver's strategy, and $q(s)$ is part of the belief system induced by the sender's strategy and the mechanism.

A key distinction is whether the mechanism makes $s_t$ effectively *cheap talk*[1]. If adding a signal does not change the feedback or payoff structure (e.g., if $F$ does not reveal outcomes and $R$ does not depend on correctness) then $s_t$ may have no direct consequences for the human's long-term payoffs. Even if $q(s)$ differs across signals, there may be little reinforcement for policies that act differently on different $s$, and the induced behavior can drift toward strategies with $\alpha(s)$ nearly constant. In equilibrium terms, pooling receiver strategies that ignore the signal become plausible.

By contrast, when the mechanism ties $s_t$ to verification options, feedback, and payoffs, ignoring differences in $q(s)$ can become costly. For example, if outcome feedback is revealed and the payoff function rewards allocating verification effort to states with low $q(s)$, then receiver strategies that condition actions on $s$ can strictly dominate strategies that do not. In this case, the uncertainty signal has operational meaning: it supports separating or near-separating regimes in which the human responds differently to different signals. The framework is thus designed to capture the idea that uncertainty in human–AI systems is not just an epistemic quantity of a model, but a strategic signal whose behavioral consequences depend on the mechanism.

### 3.3. Pooling and Near-Separating Behavioral Regimes

The signaling-game perspective allows us to formalize behavioral regimes that will be useful for characterizing human reliance. For simplicity, consider the assistant's recommendation strategy as fixed and focus on how behavior depends on the signal.

Define the *verification rate* at signal $s$ as

$$v(s) = \Pr(a_t = \text{VERIFY} \mid s_t = s),$$

and the *follow rate* as

$$f(s) = \Pr(\text{final answer} = \hat{y}_t \mid s_t = s).$$

Misuse and disuse at the signal level are

$$m(s) = \Pr(a_t = \text{ACCEPT}, \; y_t \neq \hat{y}_t \mid s_t = s),$$

$$d(s) = \Pr(\text{final answer} \neq \hat{y}_t, \; y_t = \hat{y}_t \mid s_t = s).$$

Formally, a reliance policy is *pooling* if the receiver's action probabilities are constant in the signal, for example if $\alpha(s)$, or equivalently $v(s)$, does not vary with $s$. Outcome-conditioned quantities such as misuse $m(s)$ and disuse $d(s)$ may still vary across signals because the assistant's correctness probability $q(s)$ can differ across signals; however,

---

[1]Here we use the term in the standard sense of a costless, non-binding message: the signal may be statistically correlated with correctness, but the mechanism need not provide incentives or observable reinforcement for conditioning actions on it.

such variation would not by itself reflect signal-sensitive user behavior. Behavior is *pooling-like* when action probabilities vary only weakly across $s$, even when the underlying $q(s)$ differs. In such regimes, the human acts as if the signal carries little or no action-relevant information, and verification effort is not systematically aligned with the assistant's error profile.

At the opposite extreme, a policy is *separating* if there exists a strictly increasing function $\phi$ such that $\alpha(s) = \phi(q(s))$, so that signals with higher correctness probability induce (weakly) higher acceptance rates. In practice we often only expect *near-separation*: behavior is not perfectly separating, but it still exhibits a clear monotone ordering across signals (higher-$q(s)$ signals are accepted more and verified less than lower-$q(s)$ signals). A simple and practically relevant form of separating behavior is a threshold policy in beliefs: there exists a cutoff $c \in [0, 1]$ and a subset $\mathcal{S}_{\text{low}} \subseteq \mathcal{S}$ such that $q(s) \geq c$ for $s \in \mathcal{S}_{\text{low}}$, $q(s) < c$ for $s \notin \mathcal{S}_{\text{low}}$, and verification is rare on $\mathcal{S}_{\text{low}}$ but common on its complement. In a binary high/low case, a near-separating policy would satisfy

$$v(\textsc{Low}) \ll v(\textsc{High}), \qquad m(\textsc{Low}) \ll m(\textsc{High}),$$

while maintaining low disuse overall. In such regimes, verification effort concentrates on states where the model is likely wrong, and the uncertainty signal shapes reliance.

These regimes parallel the familiar pooling and separating equilibria in signaling games: pooling behavior corresponds to receiver strategies that ignore the signal, whereas separating or near-separating behavior corresponds to strategies that respond differentially to signals in a way that tracks $q(s)$. Describing human–AI interaction in terms of $v(s)$, $f(s)$, $m(s)$, and $d(s)$ therefore provides a bridge between behavioral measurements and game-theoretic categories. This illustrates how the mechanism can determine whether uncertainty behaves as cheap talk or as a strategically meaningful signal. By making payoffs explicitly depend on correctness and by revealing feedback that allows the human to learn $q(s)$, the mechanism creates incentives for policies that condition on uncertainty and allocate verification effort to low-$q(s)$ states. In our empirical evaluation, we will instantiate such a scoring mechanism with concrete payoffs and examine the resulting behavior.

# 4. Study: Mechanisms for Uncertainty

We now instantiate the mechanisms from Section 3 in a controlled online study. The aim is to hold the AI assistant's mapping $x_t \mapsto (\hat{y}_t, s_t)$ fixed and vary only the interaction mechanism, in particular, whether uncertainty is presented as a neutral annotation or embedded in a feedback-and-scoring scheme, and to examine the resulting differences in verification, reliance, and accuracy.

## 4.1. Task, Participants, and Procedure

We recruited 180 participants via an online platform to complete a 20-item multiple-choice quiz (science, urban planning, history/geography, sustainability/energy; four options, one correct). Participants were randomly assigned between subjects to a human-only baseline, an uncertainty-only arm, or a game-theoretic arm; after removing the 1% fastest completions per arm, 178 participants remained (59/60/59), yielding 3560 trials. Participants read instructions (and, in AI-assisted arms, a description of the assistant and available actions), answered the 20 questions in random order, and we recorded completion time in all arms and, in AI-assisted arms, whether they accepted the recommendation immediately or verified before responding.

## 4.2. Mechanism Conditions

We instantiate the mechanisms discussed in Section 3.

**Baseline (human-only).** Participants saw only the question and four options, and directly selected an answer. No AI recommendation or uncertainty signal was shown, and there was no per-item feedback or scoring during the task.

**Arm 1: Uncertainty-Only Mechanism.** On each item the assistant displayed a recommended option $\hat{y}_t$ along with an uncertainty banner (high or low). Participants could either ACCEPT the recommendation and submit immediately, or VERIFY by entering a brief verification mode before committing to a final answer. This arm corresponds to an "uncertainty-only" mechanism: uncertainty is surfaced, and an explicit verify option is provided, with no per-item feedback about correctness and no scoring rule tied to behavior.

**Arm 2: Game-Theoretic Feedback Mechanism.** Here, the assistant's recommendations and uncertainty banners were identical to Arm 1. The mechanism differed only in feedback and payoffs. After each item, participants saw whether the assistant was correct, whether their final answer was correct, and their updated cumulative score. We use a simple, normalized scoring rule to induce a non-degenerate accept–verify tradeoff in a short laboratory task. The absolute scale is not meant to reflect real-world stakes; it is chosen for transparency and because any positive affine transformation would preserve the implied incentives. The key design choice is a small but nonzero verification cost, which creates a threshold in the inferred correctness probability $q$. The scoring rule was:

- If the participant accepted the assistant and the final answer was correct: $+1$ point; if wrong: $-1$ point.

- If the participant verified, they lose $-0.3$ points; after verification, a correct final answer yielded $+1$ (net $+0.7$) and an incorrect answer yielded $0$ (net $-0.3$).

This arm instantiates the scoring mechanism from Section 3. The scoring rule creates a simple tradeoff: VERIFY incurs a small fixed cost but reduces error risk, whereas ACCEPT yields a gain only when the recommendation is correct and a loss when it is not. As a result, the rational policy has a threshold form, accept when the inferred probability of correctness is sufficiently high, and otherwise verify.

Across the two AI-assisted arms, the assistant's recommendations followed a pre-scripted pattern of correctness that was balanced across items and arms. This ensures that differences in behavior or accuracy can be attributed to the mechanism, rather than to idiosyncrasies of the model. More details about the study are in Appendix A.1 and A.2.

### 4.3. Metrics

We computed the following quantities, which correspond to the behavioral summaries introduced in Section 3:

- **Verification rate**: proportion of trials with $a_t = $ VERIFY.
- **Accept rate**: proportion of trials with $a_t = $ ACCEPT.
- **Follow rate**: proportion of trials where the final answer matched the assistant's recommendation.
- **Misuse**: proportion of trials where participants accepted an incorrect recommendation without verification.
- **Disuse**: proportion of trials where participants ultimately rejected a correct recommendation.
- **Accuracy**: proportion of trials with a correct final answer.
- **Completion time**: task completion time in seconds.

### 4.4. Analysis Approach

Our primary analysis compared arm-level proportions using Wilson confidence intervals, two-proportion $z$-tests, and Fisher's exact tests. To account for repeated measures and heterogeneous item difficulty, we also fit logistic generalized linear models with two-way cluster-robust standard errors for key binary outcomes (verification, follow, accuracy).

Completion times were compared using Kruskal–Wallis tests on raw times and Welch one-way ANOVA on log-transformed times, with Mann–Whitney tests and Welch $t$-tests on log times (Holm-corrected) for pairwise contrasts.

### 4.5. Results

**Verification, Acceptance, and Following Rates.** We present the results in Figure 1. The game-theoretic mechanism increased verification relative to the uncertainty-only mechanism. In Arm 1 (uncertainty-only), participants chose to verify in 541 of 1200 trials (45.1%). In Arm 2 (game-theoretic), verification rose to 581 of 1180 trials (49.2%), a risk difference of $+4.2$ percentage points. Both the two-proportion $z$-test ($p \approx 0.042$) and Fisher's exact test ($p \approx 0.044$) indicated a significant increase, and a logistic

GLM with two-way clustered standard errors yielded an odds ratio (OR) of about 1.18 ($p \approx 0.018$). A within-item permutation test confirmed that the observed difference in verification is unlikely under a null of no arm effect (permutation $p = 0.001$).

Acceptance and following moved in the complementary direction. The accept rate decreased from 54.9% (659/1200 cases) in the uncertainty-only arm to 50.8% (599/1180 cases) in the game-theoretic arm, mirroring the verification increase (risk difference 4.2 pp; $z$-test $p \approx 0.042$, Fisher $p \approx 0.044$). The follow rate fell from 60.5% (726/1200 cases) to 55.1% (650/1180 cases), a difference of 5.4 pp ($z$-test $p \approx 0.008$, Fisher $p \approx 0.008$). The GLM for follow had an OR of 0.80 ($p < 0.001$), indicating a robust reduction in treating the assistant's recommendation as a default.

**Misuse and Disuse.** The mechanism change substantially reduced misuse without increasing disuse. Misuse was 9.8% in the uncertainty-only arm (117/1200; CI 0.082, 0.116) and 4.3% in the game-theoretic arm (51/1180; CI 0.033, 0.056), approximately halving the rate of harmful blind acceptance. Disuse (rejecting a correct recommendation) remained low and comparable across arms: 1.0% (12/1200; CI 0.006, 0.017) versus 1.1% (13/1180; CI 0.006, 0.019). Thus, the mechanism primarily suppressed confidently wrong accepts without inducing broad skepticism toward correct AI advice.

To examine whether the mechanism specifically sharpens how participants use the uncertainty signal, we further analyze behavior restricted to HIGH-uncertainty items (Fig. 2). In the uncertainty-only arm, participants already verify frequently on HIGH items (80.5%, 483/600) and follow the AI only 23.0% of the time (138/600), but the game-theoretic mechanism pushes this pattern further, increasing verification to 91.4% (539/590; $\Delta = 10.9$ pp, $p < 0.001$) and reducing follow-AI to 12.4% (73/590; $\Delta = -10.6$ pp, $p < 0.001$). For LOW-uncertainty items, verification and follow-AI rates are nearly identical across arms (all $p > 0.1$). This suggests that the mechanism does not indiscriminately increase skepticism, but reallocates human effort toward precisely those cases where the AI indicates high uncertainty, consistent with our "uncertainty as strategic signal" view.

**Accuracy.** Figure 3 shows that accuracy increased from 81.9% in the uncertainty-only arm to 85.3% in the game-theoretic arm, a risk difference of $+3.3$ pp ($z$-test $p \approx 0.028$, Fisher $p \approx 0.031$). Relative to the human-only baseline (80.3%; 948/1180 cases), the game-theoretic mechanism improved accuracy by $+4.9$ pp ($p \approx 0.002$), whereas the uncertainty-only arm did not significantly differ from baseline. A logistic GLM with two-way clustered standard errors estimated a positive but non-significant accuracy coefficient for the game-theoretic versus uncertainty-only contrast (OR $\approx 1.28$, $p \approx 0.19$). Overall, the mechanism-aware use of uncertainty yielded small but consistent gains.

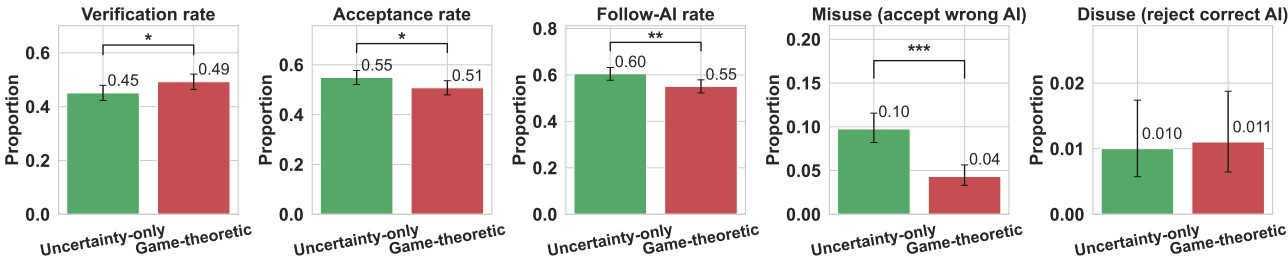

*Figure 1.* Behavioral metrics by mechanism for the two AI-assisted arms: verification, acceptance, following the AI, misuse, and disuse. Error bars denote 95% Wilson confidence intervals. Stars above brackets indicate statistically significant differences between arms based on two-proportion $z$-tests ($^*p < 0.05$, $^{**}p < 0.01$, $^{***}p < 0.001$); comparisons without stars are not significant at the 5% level.

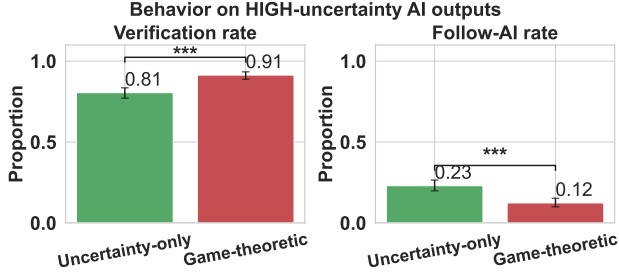

*Figure 2.* Verification and follow-AI rates on HIGH-uncertainty items for the uncertainty-only and game-theoretic mechanisms. The game-theoretic mechanism significantly increases verification ($80.5\% \rightarrow 91.4\%$) and decreases following ($23.0\% \rightarrow 12.4\%$).

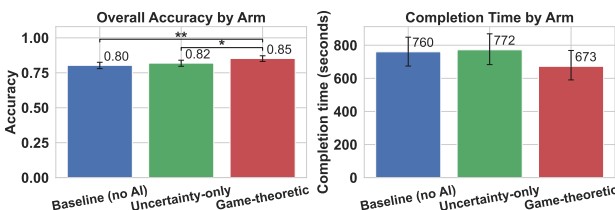

*Figure 3.* Accuracy and completion time across all three arms. The left panel reports overall accuracy with 95% Wilson confidence intervals. The right panel reports mean completion time in seconds with 95% bootstrap confidence intervals.

**Completion Time.** Mean completion times were similar across arms (baseline: 760 s; uncertainty-only: 772 s; game-theoretic: 673 s). A Kruskal–Wallis test on raw times (H = 3.29, $p \approx 0.19$) and Welch ANOVA on log times ($p \approx 0.17$) detected no significant differences. Pairwise Mann–Whitney and Welch tests on log times yielded adjusted $p$-values above 0.09 in all comparisons. Thus, differences in behavioral metrics are not explained by time-on-task.

Overall, the empirical findings align with the mechanism-aware perspective in Section 3: moving from an uncertainty-only mechanism to a feedback-and-scoring mechanism that treats uncertainty as a strategic signal increases verification, reduces harmful blind following, and improves performance while leaving time demands essentially unchanged.

## 5. Call for Mechanism-Aware Uncertainty

Our analysis suggests that uncertainty in human–AI systems should be understood at the level of the *mechanism* that links models, humans, and environment across repeated interaction. Rather than treating uncertainty as a static property of model outputs, evaluated in isolation or under implicit decision rules, we have argued that its significance lies in how it shapes policies of reliance once signals, actions, feedback, and payoffs are fixed. This section develops that perspective and clarifies what it implies for modeling and system design.

### 5.1. Mechanisms as the Unit of Analysis

The framework in Section 3 treats a human–AI system as a repeated interaction governed by a mechanism that specifies which signals the assistant may emit, which actions are available, what feedback is revealed, and how rewards accumulate over time. Given such a mechanism, uncertainty is a strategic signal whose effect is determined by this coupling, including how often ground truth enters the loop through naturally labeled instances or occasional "probe" cases with known answers. The empirical comparison in Section 4 shows that changing only the mechanism around a fixed sequence of $(\hat{y}_t, s_t)$ is enough to alter verification, misuse, and accuracy, so analyses of uncertainty that ignore mechanisms can miss important determinants of behavior and undervalue the role of the surrounding interaction design.

### 5.2. Policies of Reliance and Behavioral Regimes

A mechanism-aware view puts *policies of reliance* at the center. Users interact with an assistant repeatedly, observe outcomes when available, and gradually adopt heuristics about when to rely and when to verify. In our formulation, these heuristics are policies that map signals and histories to actions. Uncertainty is valuable when it shapes policies that concentrate verification where it is needed and avoid unnecessary checking where the assistant is reliably correct. Simple uncertainty-only mechanisms can already support such selective reliance, as our results illustrate, but their effect is mediated by the mechanism in which they are

embedded. When feedback is sparse, these policies are updated only on interactions where ground truth is known, whether because the task naturally yields labels or because the system occasionally surfaces known-answer cases.

Within this perspective, notions such as pooling and separating acquire a concrete behavioral meaning. In a pooling-like regime, the verification rate $v(s)$ and follow rate $f(s)$ depend weakly on the signal: high- and low-uncertainty recommendations are treated similarly, and the misuse rate $m(s)$ can be substantial even when signals are informative. In a near-separating regime, verification is concentrated on signal states associated with higher error, while low-uncertainty states are mostly accepted; misuse is low where $q(s)$ is high, and disuse is kept small. Misuse and disuse summarize how far a mechanism is from an ideal in which uncertainty drives targeted verification.

### 5.3. System-Level Design of Human–AI Mechanisms

Mechanisms operate at the system level: they determine which actions are available and how they are valued, while leaving the underlying model and its uncertainty fixed.

Many deployed systems implicitly resemble an uncertainty-only mechanism. Predictions are accompanied by uncertainty scores or hedged language, but the surrounding interaction is not responsive to them: users lack structured opportunities to verify, outcome feedback is sparse or informal, and incentives do not distinguish careful from careless use of AI assistance. **In real-world settings where ground truth is often unavailable at decision time, this does not imply that feedback must be abandoned; it implies that feedback must be designed.** Systems can reserve a small share of interaction for decisions with known or easily recoverable outcomes, for example by revisiting cases whose resolution is already known or periodically presenting deliberately constructed probe instances, and use these to provide targeted feedback and recalibrate reliance.

**Call for action.** For system builders, the main implication is that uncertainty and mechanisms should be treated as a single design surface rather than as separate concerns. Choosing a representation for uncertainty is only one part of the problem. Equally important are questions such as:

- *Exposure:* when and how is uncertainty surfaced to users, and does its presentation invite different actions rather than merely decorate predictions?
- *Action set:* what concrete options are available in response to uncertain recommendations (verify, escalate, ignore), and are these options easy to exercise within existing workflows?
- *Feedback:* on which subset of decisions will outcomes eventually be known, and how are these observations fed back so that policies of reliance can adapt over time?

- *Probing and calibration:* can the system occasionally initiate decisions on cases with known or easily recoverable labels, in order to probe current behavior and maintain calibration between humans and models?

In general, treating mechanisms as first-class design objects suggests concrete levers beyond model-side improvements, such as reserving a small labeled subset of cases for systematic feedback, tying evaluation and incentives to how uncertainty is *used* rather than to accuracy alone, or selectively amplifying or suppressing uncertainty displays where they help or hinder near-separating policies of reliance. We do not claim that these choices are universally preferable, but they illustrate how mechanism-level decisions can shape reliance behavior even when the model is unchanged.

## 6. Alternative Views

Our position sits alongside several coherent alternatives. One views uncertainty primarily as a model-centric reliability and abstention problem: the aim is to produce well-calibrated uncertainty scores and selective-prediction policies that abstain on high-uncertainty cases, assuming human decision-makers or downstream procedures will then use these signals appropriately (Guo et al., 2017; Geifman & El-Yaniv, 2017). Another emphasizes cognitive and interaction factors: patterns of overreliance and underreliance are attributed to automation bias, mental models, and interface design, so interventions focus on explanations, cognitive forcing functions, and other UX techniques rather than explicit incentives (Buçinca et al., 2021; Bansal et al., 2021; Alon-Barkat & Busuioc, 2023). A further perspective is skeptical of applying mechanism-design ideas to human–AI interaction at all, citing violated utility assumptions, ethical or organizational concerns about incentive schemes, and a preference for simpler informational interfaces guided by professional norms (Alon-Barkat & Busuioc, 2023; Green & Chen, 2019). We view these perspectives as complementary to our own: calibration, abstention, and cognitive factors are clearly important. Our contribution is to argue that, in practice, uncertainty is always embedded in some mechanism, explicit or implicit, that shapes payoffs and feedback, and that making this mechanism explicit helps explain both why technically sound uncertainty estimates and carefully designed interfaces can sometimes still yield pooling-like behavior and how mechanism choices can amplify the impact of uncertainty when it is already being used.

## 7. Conclusion

We propose a game-theoretic view of human–AI decision support in which uncertainty is treated as a strategic signal within a repeated mechanism, shaping policies of reliance that can resemble pooling or near-separating regimes. In a

pilot study, a game-theoretic mechanism with explicit verification, feedback, and incentives increased verification and reduced harmful misuse relative to an uncertainty-only interface, showing how mechanism-level design choices around uncertainty can influence reliance in human–AI systems.

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

# A. More on the Study

## A.1. Questions Generation

The main study used a 20-item multiple-choice knowledge quiz with short text questions and four answer options per item, as described in Section 4. Items were constructed to be answerable by lay adults without specialized training and to support tight control over correctness and timing.

We defined four topical domains that are representative of common information-seeking and decision-support use cases and admit objectively verifiable answers: (i) science, (ii) urban planning, (iii) history and geography, and (iv) sustainability and energy. For each domain we drafted an initial pool of candidate questions under the following constraints:

- exactly one unambiguously correct option and three plausible but incorrect distractors;

- short, self-contained questions;

- answers that can be checked against reliable public references (e.g., encyclopedic or governmental sources);

- a mix of easier and harder items to avoid ceiling or floor effects in unaided human performance.

From this pool, we first filtered out ambiguous or poorly discriminating questions via internal pilot testing, then selected a final set of 20 items that (i) balanced topical coverage across the four domains and (ii) avoided overlapping content or repeated facts. The final question set was fixed prior to running the experiment.

During the study, participants saw the same 20 items in a uniformly random order, with item order independently randomized for each participant. Question wording, correct answers, and distractors were held fixed across all conditions. Only the presence or absence of the assistant's recommendation and uncertainty banner, and the surrounding mechanism (feedback and scoring), varied by arm.

## A.2. Model and Uncertainty Measurement Process

In our framework in Section 3, the assistant acts as a sender that maps each instance $x_t$ to a recommendation $\hat{y}_t$ and an uncertainty signal $s_t \in \mathcal{S}$. In the experiment, we instantiate this assistant using an LLM (Llama 3.1-8B-Instruct) that answers each question and provides a binary self-report that we map into a binary *uncertainty* signal.

**LLM-Based Assistant.** Concretely, for each multiple-choice item we present the LLM with the question stem and four labeled options (A–D) in plain text. We then issue a single prompt that instructs the model to:

1. choose exactly one option (A, B, C, or D) as its final answer, and

2. state whether it is confident that this answer is correct, using a forced-choice binary response (Yes or No).

A schematic prompt template is:

```
You are answering a multiple-choice quiz.
Question: <question text>
Options: A) <text>; B) <text>; C) <text>; D) <text>
First, state your final answer as one letter (A, B, C, or D).
Then answer this question: "Are you confident that your answer is correct?"
Respond with:
Answer: <A/B/C/D>
Confident: <Yes/No>
```

From the model's response we extract:

- the recommended option $\hat{y}_t \in \{A, B, C, D\}$, and

- a binary self-reported confidence label $c_t \in \{\text{YES}, \text{NO}\}$.

We then define the uncertainty signal $s_t$ by mapping (so that HIGH corresponds to *higher uncertainty*):

$$s_t = \begin{cases} \text{LOW} & \text{if } c_t = \text{YES}, \\ \text{HIGH} & \text{if } c_t = \text{NO}. \end{cases}$$

Ground-truth correctness is computed by comparing $\hat{y}_t$ to the known correct option for each item.

**Balancing Correctness and Uncertainty.**    To ensure that the uncertainty signal is both informative and non-trivial, we aim for a setting in which the assistant is neither nearly always correct nor nearly always low-uncertainty. Practically, this is achieved in two steps.

First, we run the LLM once on each question in a larger candidate pool (using the prompt above) and record, for each item, whether the model's answer is correct and whether it self-reports being confident. This yields four possible outcome types, where LOW/HIGH refer to *uncertainty*:

1. correct and low-uncertainty (C/L),

2. correct and high-uncertainty (C/H),

3. incorrect and low-uncertainty (W/L),

4. incorrect and high-uncertainty (W/H).

Second, from this pool we select a subset of 20 items such that:

- approximately half of the items are answered correctly and half incorrectly by the model, and

- both uncertainty levels (HIGH and LOW) are well represented.

On the realized 20-item set, the assistant is correct on $8/10$ LOW-uncertainty items and $5/10$ HIGH-uncertainty items, i.e., $q(\text{LOW}) = 0.80$ and $q(\text{HIGH}) = 0.50$ ($\Delta = 0.30$). This yields a non-trivial setting in which uncertainty is informative about correctness while still leaving substantial room for both beneficial verification and harmful blind acceptance.

**Fixing the Assistant Across Mechanisms.**    Once the mapping $x_t \mapsto (\hat{y}_t, s_t)$ has been obtained from the LLM for the selected 20 items, we treat it as a fixed assistant for the purposes of the main experiment. No further model queries are made during the study, and the assistant does not adapt over time.

This design keeps the assistant grounded in a realistic LLM-based answering process with a binary uncertainty signal (derived from a binary self-report), while holding the sequence of $(\hat{y}_t, s_t)$ pairs fixed across mechanisms. As a result, any differences in verification behavior, misuse, disuse, or accuracy between the uncertainty-only and game-theoretic arms can be attributed to the surrounding mechanism $M = (\mathcal{S}, \mathcal{A}, F, R)$ rather than to changes in the underlying model.

### A.3. Participant Demographics

For the study, we used Prolific as the platform to recruit the participants, as it supports fair participant compensation. We recruited 180 participants aged from 19 to 79 years old, with a mean age of 36.24 years and a standard deviation of about 13.01 years. In terms of gender, 48.20% of participants identified as male and 38.85% as female. The remaining 12.95% were not willing to share their gender. Overall, the sample includes participants from 29 different countries of residence distributed across multiple continents.

### A.4. Limitations and Scope

Our pilot empirical study is intended as an initial illustration of the framework rather than a comprehensive evaluation across domains, and it has several limitations.

First, the task is a multiple-choice knowledge quiz with short text questions and four options. This setting allows tight control over correctness and timing, but it differs substantially from many high-stakes decision-support contexts in which

model assistance is deployed (e.g., clinical diagnosis, legal analysis, or policy design). In such domains, information is less neatly structured, costs and benefits are more asymmetric, and verification may require expertise or coordination with other actors. The behavioral patterns we observe should therefore be interpreted as evidence that mechanisms can sharpen separation between low- and high-uncertainty regimes in a stylized environment, starting from an uncertainty-only baseline that already induced strongly uncertainty-sensitive behavior, rather than as direct estimates of effects in any particular application area.

Second, we study a single assistant with a simple, binary uncertainty signal. Holding the assistant fixed across mechanisms isolates the role of the mechanism $M$, but it also narrows the space of possible dynamics. Different models, richer signal spaces, or alternative mappings from internal states to displayed uncertainty could change how humans form $q(s)$ and how easily they can learn to condition on uncertainty, potentially either weakening or amplifying the kinds of separation we observe. Similarly, we focus on a specific family of verification actions and do not explore other forms of human agency (such as querying for explanations, asking the model to reconsider, or escalating to another expert).

Third, the mechanisms we compare use a particular scoring rule and a specific implementation of outcome feedback. Our analysis of myopic best responses relies on approximate risk neutrality and on the assumption that verification is usually successful when the assistant is wrong. Real-world deployments may involve different incentive schemes, intrinsic motivations, and risk attitudes, as well as verification processes that are noisy or constrained. We do not claim that our scoring rule is optimal or uniquely justified; it serves as a concrete instance of a mechanism that ties uncertainty to feedback and payoffs in a way that makes threshold-like reliance policies individually attractive and can further concentrate verification on high-uncertainty cases.

