# OpenReview forum: "Position: Uncertainty is a Strategic Signal in Human–AI Decision Making"
_ICML.cc/2026/Position_Paper_Track — ICML 2026 Position Paper Track regular_

### Official Review · Reviewer_ZRP4 · 2026-03-02

**Significance:** 3
**Argument Clarity:** 3
**Rating:** 4
**Confidence:** 3

**Questions:**

1. Does the paper have references supporting the background assumption that AI systems report uncertainty in a consistent manner, such that q(s) can be treated as a fixed function? I would appreciate clarification on this assumption.

2. I am confused by the finding that task completion time in the game-theoretic condition is faster than in the human-only baseline. Given that users in the game-theoretic condition must additionally process the uncertainty signal, evaluate whether to accept or verify, and engage with the feedback and scoring after each item, I would expect this cognitive overhead to result in longer completion times compared to users making independent decisions without any AI uncertainty influence. Could the authors give more explanation?

3.  In the regime analysis, the authors briefly characterize the two extreme cases of pooling and separating, with other cases described as lying in a near-separating regime. I find this analysis to be qualitative rather than quantitative. Specifically, I would like to know how v(s), f(s), m(s), and d(s) vary with respect to $\alpha(s)$. What quantitative evidence supports the claims that v(LOW) $\ll$ v(HIGH) and m(LOW) $\ll$ m(HIGH)?

**Alternative Views Section:**

Yes

**Compliance With Llm Reviewing Policy A Conservative:**

Affirmed.

**Discussion Potential:**

2

**Final Justification:**

My concerns have been adequately addressed. I am raising my score to 4.

**Paper Summary:**

This paper argues, in assisted decision making, the AI reported uncertainty should be treated as a strategic signal rather than a static model property or purely informational label. The authors formalize a framework in which the interface specifies uncertainty signals, user responses, and the resulting policy-shaping consequences, using these repeated steps to characterize near-separating reliance regimes. The authors use pilot study with 180 participants to  support their proposition.

**Position:**

No

**Position In Title:**

Yes

**Related Work:**

2

**Strengths And Weaknesses:**

# Strength
The most inspired result of this work, in my view, is that the well-desinged human study reveals that moving from an uncertainty-only mechanism to a feedback-and-scoring mechanism that treats uncertainty as a strategic signal increases verification, reduces harmful blind following, and improves performance while leaving time demands essentially unchanged. I consider this a valuable and actionable design direction for uncertainty mechanism design in future AI-assisted decision-making systems.

# Weakness

1. I feel this paper reads more as an exposition of how we should view and use uncertainty in AI-assisted decision making than as a strong position paper. The reason it lacks a strong position is the absence of a genuinely opposed view. Existing work that treats AI-assisted decision making as single-round and does not consider how reliance changes over time simply operates under different scenarios from the present work. These are not contradictory positions, and I do not consider them to be in conflict with the authors' framework.

2. The alternative views in a position paper should describe and address one or more credible positions that are genuinely opposed to the paper's position. However, the alternative views the authors discuss are, by their own admission, complementary to their perspective rather than opposed to it. This does not meet the standard expectation for what alternative views should present. I think this issue is closely connected to Weakness 1: both stem from the fact that this work does not occupy a position that conflicts with existing work, but rather offers insights into the regime analysis when human reliance on AI-reported uncertainty changes over time.

3. In human study, mechanism parameters are fixed without justification, i.e., the scoring rule parameters. The experiments do not verify the generalizability of the conclusions across different parameter designs, nor do the authors analyze whether the feedback-and-scoring mechanism retains its advantage over the baselines independently of the specific parameter choices made.

**Support:**

3

---

> ### Author Rebuttal · Authors · 2026-03-30
>
> We appreciate the reviewer’s comments on the paper’s positioning. We will make a significant effort to address this in the manuscript and welcome the opportunity to clarify our position and the opposing one here.
>
> ## Response to W1/2
> The opposing view we intend to identify treats uncertainty as a model-side property or informational label presented to the user in isolation, without taking the surrounding interaction mechanism and the evolution of reliance over time as the main object of analysis. Our position is that, in human–AI decision support, uncertainty should instead be treated as part of the interaction mechanism through which signals, actions, feedback, and consequences shape reliance policies over repeated use.
>
> Because this position is relatively new, we described prior work as complementary in the sense that it provides important ingredients for the perspective we advocate. However, this wording may have obscured the actual point of opposition. The contrast is not merely between different scenarios, but between two ways of framing uncertainty: as a property or label attached to a prediction, or as a strategic signal embedded in a repeated mechanism. From that perspective, our paper does take a genuine position.
>
> Our perspective is reflected not only conceptually in the framework and alternative views sections, but also **empirically** in the user study, where we compare uncertainty presented alone against uncertainty embedded in a game-theoretic mechanism and show that the latter changes reliance behavior. We are committed to revise the wording, especially in the related work and alternative-views discussion, to make this opposition clearer, in light of the reviewer's feedback
>
> ## Response to W3
> The study intentionally uses a simple normalized scoring rule to instantiate one transparent accept/verify tradeoff while holding the assistant’s outputs fixed across arms. As stated in Section 4.2, the absolute scale is not meant to represent real-world stakes; the key design feature is a small but nonzero verification cost, which creates a threshold in inferred correctness probability and a non-degenerate reliance problem in a short pilot study.
>
> Our claim is not that the empirical advantage is invariant to every parameterization. Rather, it is that a concrete mechanism with feedback and payoff consequences can change reliance behavior even when the underlying (ŷ_t, s_t) sequence is held fixed across arms. Studying robustness across alternative parameterizations is an important next step, and we will emphasize this in the main paper.
>
> ## Response to Q1
> Our framework does not assume that all AI systems report uncertainty in a globally stable way. In Section 3, q(s) is the human’s subjective belief about correctness conditional on the signal for a fixed assistant and mechanism; it can only be learned from sparse feedback and may depend on deployment context. In the empirical study, this assumption is enforced by design: the assistant’s mapping x_t ↦ (ŷ_t, s_t) is precomputed once and then held fixed across mechanisms, so behavioral differences can be attributed to the mechanism rather than signal drift.
>
> ## Response to Q2
> We do not interpret Figure 3 as evidence that the game-theoretic condition is faster than the human-only baseline. The paper’s actual statistical result is that **completion times are not significantly different across arms**: Kruskal–Wallis on raw times p≈0.19, Welch ANOVA on log times p≈0.17, and pairwise adjusted tests all above 0.09. The intended takeaway is only that the added mechanism is not associated with a detectable increase in time-on-task in this study.
>
> ## Response to Q3
> Our intention in Section 3.3 is to introduce pooling and near-separating regimes as behaviorally measurable categories, rather than to fit a full structural equilibrium model. The paper already defines v(s), f(s), m(s), d(s), and the empirical study provides quantitative evidence for near-separation in the binary case: the signal is informative, with q("LOW")=0.80 and q("HIGH")=0.50; and behavior differs systematically by signal. In particular, on HIGH items verification rises from 80.5% to 91.4% and follow-AI falls from 23.0% to 12.4%, while LOW-item behavior is nearly unchanged across arms. At the aggregate level, misuse is reduced from 9.8% to 4.3% without increasing disuse.
>
> More broadly, our goal in this position paper is to support the conceptual claim with initial empirical evidence: namely, that treating uncertainty at the level of the interaction mechanism reveals behaviorally meaningful differences that can already be observed in practice. We agree that a fuller characterization of these regimes would be valuable, and we hope this paper helps open that discussion.
>
> We hope this clarifies both our position and the existence of an opposing view, even if the current wording did not yet make the contrast sufficiently explicit, which we are committed to address in our revised manuscript.

---

> > ### Author Rebuttal · Reviewer_ZRP4 · 2026-04-05
> >
> > Thanks for your clarification. Given that my concerns have been adequately addressed, I am raising my score to 4.

---

### Official Review · Reviewer_q9DG · 2026-03-02

**Significance:** 3
**Argument Clarity:** 3
**Rating:** 4
**Confidence:** 3

**Questions:**

See Weaknesses

**Alternative Views Section:**

Yes

**Compliance With Llm Reviewing Policy A Conservative:**

Affirmed.

**Discussion Potential:**

3

**Final Justification:**

After the rebuttal, my concerns were resolved. I recommended that the authors incorporate those changes/clarifications as they promised. I keep my overall positive score.

**Paper Summary:**

The paper argues that uncertainty quantification approaches should be embedded, understood, and studied within a concrete deployment mechanism rather than treated as a property of the model itself.

They provide an empirical study showing that, when the end user acts solely on the provided uncertainty estimate, without a useful feedback signal, the end user is often under- or over-reliant on those predicted measures. In contrast, when, in addition to the uncertainty estimate, a feedback signal is provided (which takes into account the history and how often the AI assistant was correct), the end user becomes more selective, often verifies answers, reduces misuse, and improves over other reliability metrics.

The authors call on the community not only to choose the representation of uncertainty measures alone, but also to consider a mechanism for how the end user will interact with the uncertainty-aware system.

**Position:**

Yes

**Position In Title:**

Yes

**Related Work:**

3

**Strengths And Weaknesses:**

## Strengths:

1) The paper is well-written and easy to follow. The authors' position is clear.
2) The authors conducted a real-world experiment/survey, where they demonstrated significant statistical differences for different groups of respondents, depending on the mechanism of the uncertainty-aware system they interacted with.

## Weaknesses:

1) I feel that an important baseline is missing in the authors' comparisons to make conclusions. Specifically, is the uncertainty measure even useful to display when feedback (e.g., how often the AI assistant was correct) is provided?
To me, it seems that only feedback itself may be the key component. And if the feedback is presented to the end-user (without the HIGH/LOW uncertainty banner), the user can improve those metrics. Would it be possible to conduct the experiment?

2) I am unsure about the results for Completion time (Figure 3). How is it possible that the completion time decreased for the Game-theoretic mechanism, given that the verification rate increased? Are those groups, used in the experiment, "statistically equivalent"? Otherwise, I do not understand why they complete the task faster, given that they verify more often.

3) What is unclear to me is the choice of the threshold of the HIGH/LOW, and this potentially could result in unfair comparison. Could the uncertainty threshold be initially chosen incorrectly? Therefore, those who did not receive explicit feedback (the uncertainty-only group) were "misled"? While people who received feedback had the opportunity to effectively "adjust" the threshold, using the feedback provided.
What could be interesting (and is currently missing) is to consider another group that follows the same protocol as the "Uncertainty-only" group, but with the HIGH/LOW threshold chosen from the "Game-theoretic" group behaviour.
This experiment would exclude the incorrect threshold choice, and in this case, we can fairly compare the "Uncertainty-only" group vs. the "Game-theoretic" group.

**Support:**

3

---

> ### Author Rebuttal · Authors · 2026-03-30
>
> We appreciate the reviewer’s careful reading and constructive comments. Below, we address the weaknesses and questions raised in the review.
>
> ## Response to W1
> On the reviewer’s concern about the missing feedback-only baseline, we would like to clarify that the present results already suggest that the uncertainty signal remains behaviorally active under the game-theoretic mechanism: the largest behavioral changes are concentrated on HIGH-uncertainty items, where verification increases from 80.5% to 91.4% and follow-AI decreases from 23.0% to 12.4%, while behavior on LOW-uncertainty items is nearly unchanged across arms. This pattern is more consistent with feedback sharpening how users act on the uncertainty signal, rather than simply inducing a signal-agnostic change in behavior. So the uncertainty alone effect is present. At the same time, we agree, that a feedback-only arm would be a valuable additional control for separating these explanations more cleanly. More broadly, to the best of our knowledge, this position paper is among the first to argue for moving beyond a display-only, memoryless use of uncertainty toward a repeated, game-theoretic mechanism. As such, we view feedback-only baselines, richer evaluations aligned with high-stakes domains, and related extensions as important next steps that this paper aims to motivate, and we will highlight this more explicitly in the main paper.
>
> ## Response to W2
> On the reviewer’s concern about completion time, we do not interpret Figure 3 as evidence that the game-theoretic mechanism makes participants faster. Participants were randomly assigned, and the time differences across arms are not statistically significant: the paper reports a Kruskal–Wallis test on raw times with p ≈ 0.19, a Welch ANOVA on log times with p ≈ 0.17, and pairwise adjusted tests above 0.09. Our intended takeaway is therefore only that the increase in verification is not accompanied by a detectable increase in time-on-task, not that the mechanism speeds users up. We will clarify this point more explicitly in the main paper.
>
> ## Response to W3
> On the reviewer’s concern about the HIGH/LOW threshold, the thresholding was fixed before the main experiment and held identical across the two AI-assisted arms. In the selected 20-item set, the assistant is correct on 8/10 LOW items and 5/10 HIGH items, so the two signals already differ meaningfully in correctness probability (q(LOW)=0.80, q(HIGH)=0.50). The game-theoretic group therefore does not benefit from a different or behavior-derived threshold; rather, both groups see the same fixed sequence of (ŷ_t, s_t), and the difference is only the surrounding mechanism. In that sense, the game-theoretic arm is not “adjusting the threshold,” but learning a different reliance policy in response to the same signal. We agree that sensitivity to alternative thresholding rules would be interesting, and we will clarify more explicitly in the main paper why the current fixed-signal design supports a fair comparison between the two arms.
>
> We would like to thank the reviewer once again for the constructive feedback and for the positive assessment of our paper. We hope the above clarifications have addressed the main concerns and made our contribution scope clearer.

---

> > ### Author Rebuttal · Reviewer_q9DG · 2026-04-03
> >
> > My concerns were resolved. Please incorporate those changes / clarifications as promised. I keep my score.

---

### Official Review · Reviewer_o6uY · 2026-03-04

**Significance:** 3
**Argument Clarity:** 3
**Rating:** 5
**Confidence:** 3

**Questions:**

- Q1: How can the framework maintain a separating regime in environments where ground-truth feedback is delayed or only available for a tiny fraction of cases?


- Q2: In a truly repeated setting, how would the mechanism account for a human user gaming the AI signal or an AI model that adaptively changes its confidence mapping based on human feedback?

**Alternative Views Section:**

Yes

**Compliance With Llm Reviewing Policy A Conservative:**

Affirmed.

**Discussion Potential:**

3

**Final Justification:**

The authors’ rebuttal has adequately addressed my concerns, accordingly, I have raised my score and support the acceptance of this manuscript.

**Paper Summary:**

This paper advocates for viewing human-AI decision support as a repeated game-theoretic mechanism where AI uncertainty serves as a strategic signal to shape user reliance policies over time. It contributes a formal framework and empirical evidence demonstrating that embedding uncertainty within an incentive-aware mechanism significantly reduces harmful overreliance and improves decision accuracy compared to static information displays.

**Position:**

Yes

**Position In Title:**

Yes

**Related Work:**

3

**Strengths And Weaknesses:**

Strengths

- S1: The paper successfully transitions the study of AI uncertainty from a static UI visualization problem to a dynamic, mechanism-design challenge. By framing uncertainty as a strategic signal rather than a mere label, it addresses why technically sound uncertainty displays often fail to change user behavior in practice.

- S2: The 180-participant pilot study offers clear, statistically significant evidence that embedding uncertainty within an incentive-aware mechanism (feedback + scoring) reduces misuse (blind following) by nearly half and improves overall decision accuracy without increasing time-on-task.



Weaknesses

- W1: The reliance on a 20-item multiple-choice quiz limits the ecological validity of the findings. High-stakes professional domains (e.g., medical diagnosis or legal analysis) involve far more asymmetric costs, unstructured data, and complex verification processes that the simple scoring rule may not capture.

- W2: The proposed mechanism's effectiveness hinges on immediate ground-truth feedback, which is frequently unavailable or prohibitively expensive in real-world AI deployments. While the authors suggest probing with known cases, the scalability of this approach remains underexplored.

**Support:**

3

---

> ### Author Rebuttal · Authors · 2026-03-30
>
> We are thankful to the reviewer for the positive assessment and thoughtful questions. We address the noted weaknesses and questions below.
>
> ## Response to W1
> On the reviewer’s concern about ecological validity, we agree that the current experiment uses a stylized pilot setting, as already noted in the paper’s limitations. At the same time, we believe this choice is useful in the present study because it allows the mechanism-level effect to be isolated under controlled conditions. Since this is one of the first empirical instantiation of our position, the 20-item quiz serves as a minimal testbed sufficient to corroborate the core claim that a repeated, game-theoretic mechanism can shape reliance differently from a display-only uncertainty interface. In more open-ended settings, additional factors can affect reliance behavior and make that effect harder to identify cleanly. However, the broader framework is intended to extend beyond this specific task structure: in more complex domains, the same design perspective applies, but with domain-specific action spaces, verification costs, and payoff asymmetries. We are committed to clarify this motivation and scope more explicitly in the main paper.
>
> ## Response to W2
> On the reviewer’s concern about the need for immediate feedback, the current experiment indeed uses immediate feedback as part of its specific empirical instantiation, since this makes the mechanism’s effect observable over a short study horizon. The broader framework, however, does not require immediate feedback on every case. What it requires is that, over repeated interaction, users encounter some consequence structure that differs across high- and low-uncertainty decisions. Concretely, this can come from delayed outcomes that become known later, selective review of a subset of cases, or occasional known-answer / probe cases used for calibration. In such settings, feedback may be sparse or delayed, but it can still shape reliance behavior over time if it is systematically linked to how uncertainty is acted upon. We will clarify this distinction more explicitly in the main paper.
>
> ## Response to Q1
> On the reviewer’s question about maintaining separation under sparse or delayed feedback, this is an important issue and one we already try to highlight in the paper’s discussion of sparse feedback and probe cases. A separating regime does not require immediate feedback on every case. It requires that, over repeated interaction, acting differently on high- versus low-uncertainty cases leads to different expected consequences. In practice, this can still be sustained through delayed outcome feedback, selective audits, or probe cases, so that high-uncertainty cases retain a higher expected value of verification than blind acceptance. Sparse feedback will generally make the effect slower or weaker, but it does not remove the basic mechanism. We will make sure to clarify this point more concretely in the main paper.
>
> ## Response to Q2
> On the reviewer’s question about strategic user behavior and adaptive confidence remapping, these are important extensions of the repeated-setting perspective. In the current study, we intentionally keep the assistant and its uncertainty mapping fixed in order to isolate the mechanism-level effect. In a real repeated deployment, strategic user behavior can be addressed through mechanism choices such as selective audits, occasional probe cases, and audit/feedback rules that make systematic blind acceptance of high-uncertainty outputs unattractive over time. Likewise, if the AI system adaptively changes its confidence mapping, the mechanism should be paired with ongoing calibration monitoring and periodic updating, since the quality of the signal is one of the mechanism’s operating assumptions. We will clarify in the main paper that the current experiment fixes the signal for identification, while strategic user adaptation and adaptive confidence remapping are natural extensions of the framework.
>
> We thank the reviewer again for the thoughtful feedback and hope that our clarifications have addressed the main concerns.

---

> > ### Author Rebuttal · Reviewer_o6uY · 2026-04-04
> >
> > Thank the authors for their rebuttal. I find their explanations convincing and will accordingly raise my score.

---

### Official Review · Reviewer_qKmd · 2026-03-08

**Significance:** 3
**Argument Clarity:** 3
**Rating:** 4
**Confidence:** 3

**Questions:**

The scoring rule is based on explicit incentives that are not common in many real-world deployments. How would this mechanism-level design work in settings where scoring users is ethically not feasible, e.g., clinical decision support?

**Alternative Views Section:**

Yes

**Compliance With Llm Reviewing Policy A Conservative:**

Affirmed.

**Discussion Potential:**

4

**Paper Summary:**

The paper proposes a mechanism-design approach to the uncertainty output of human–AI decision-making systems, in contrast to just displaying the uncertainty as info label. Based on signaling game theory, the paper puts forward a framework where a mechanism, defined by signal space, feedback, and payoff function, controls whether uncertainty causes "pooling" or "separating" behavior.

The paper conducts a pilot study with 180 between-subjects and compares three mechanisms: human-only baseline, uncertainty-only AI interface, and a game-theoretic AI interface with per-item feedback and a scoring rule. The proposed game-theoretic approach increases verification rates, reduces harmful misuse, and improves accuracy compared to the other mechanism.

Call for action: The paper calls on system designers to treat uncertainty and interaction mechanisms as a joint design surface.

**Position:**

Yes

**Position In Title:**

Yes

**Related Work:**

3

**Strengths And Weaknesses:**

**Strengths**

1. The proposed approach is of real importance to the community, since AI-based assistants are being deployed in serious settings. The mechanism-design proposal is novel.

2. The paper's main idea, namely, that uncertainty should be treated as a mechanism-level strategic signal rather than an epistemic quantity of a model, drives the formalism and the empirical work. As a result, this is a coherent conceptual contribution.

3. The paper discusses alternative approaches: model-centric reliability and abstention problem, cognitive and interaction factors, and skepticism toward mechanism-design approaches. The paper considers these alternatives as complementary.

**Weaknesses**

The paper's position aims for high-stakes domains such as medicine or finance, but the empirical support is based on a 20 item multiple-choice  quiz. Although the paper hints this issue in the appendix, this could have been addressed in the main body.

**Support:**

3

---

> ### Author Rebuttal · Authors · 2026-03-30
>
> We appreciate the reviewer’s constructive comments and the opportunity to clarify the points raised in the review.
>
> ## Response to W1
> Thank you for pointing this out. We will revise the main paper to state explicitly and furhter highlight that the experiment is a pilot study based on a 20-question setting. This concern is also addressed more fully in our response to reviewer o6uY, Response to W1.
>
> ## Response to Q1
> We fully agree that many real deployments should not literally assign scores to users. In our framework, the scoring rule is just a transparent experimental vehicle for instantiating a more general design principle. The broader mechanism is defined by three elements: the available actions, the feedback users receive, and the consequences attached to decisions over repeated interaction. In clinical decision support, for example, this need not take the form of explicit incentives: high-uncertainty cases may require escalation or mandatory second review, and feedback may be delivered retrospectively as soon as outcomes become known. This reinforces the fact that we consider a longer-term interaction mechanism rather than isolated human-AI interactions. We thank the reviewer for highlighting this, and we will clarify further that the substantive claim of the paper is not that practitioners should “score users,” but that uncertainty should be designed and evaluated together with the surrounding decision mechanism. We will make this distinction explicit in the main paper.
>
> We thank the reviewer again for the thoughtful feedback and hope the above clarifications have addressed the main concerns and made our intended scope clearer.

---

> > ### Author Rebuttal · Reviewer_qKmd · 2026-04-04
> >
> > I thank authors for addressing my questions

---

### Decision · Program_Chairs · 2026-04-30

**Decision:**

Accept (regular)

**Comment:**

Reviewers are mildly positive about the paper. The paper indeed contains some merits backed up by the pilot study, while the evidence and the alternative views could be made stronger. Overall this would be a nice addition to ICML, while rejecting it probably won't be a big loss.